# Near-Infrared Absorption Properties of Neutral Bis(1,2-dithiolene) Platinum(II) Complexes Using Density Functional Theory

**DOI:** 10.3390/nano12101704

**Published:** 2022-05-17

**Authors:** Xuan-Hoang Luong, Nguyet N. T. Pham, Kyoung-Lyong An, Seong Uk Lee, Shi Surk Kim, Jong S. Park, Seung Geol Lee

**Affiliations:** 1School of Chemical Engineering, Pusan National University, 2, Busandaehak-ro 63 beon-gil, Geumjeong-gu, Busan 46241, Korea; xuanhoangluong@pusan.ac.kr; 2Faculty of Chemistry, University of Science, Vietnam National University Ho Chi Minh City, Ho Chi Minh City 721337, Vietnam; 3NANOCMS Co., Ltd., 48, 4sandan 4-ro, Jiksan-eup, Seobuk-gu, Cheonan-si 31040, Korea; ankl@nanocms.co.kr (K.-L.A.); lsw4004@nanocms.co.kr (S.U.L.); franz@nanocms.co.kr (S.S.K.); 4Department of Organic Material Science and Engineering, Pusan National University, 2, Busandaehak-ro 63 beon-gil, Geumjeong-gu, Busan 46241, Korea

**Keywords:** platinum complex, dithiolene complex, NIR, density functional theory, GW-BSE approximation

## Abstract

Small metal complexes are highly interesting for bioimaging because of their excellent near-infrared (NIR) absorption properties. In this study, neutral complexes of platinum(II) connected to two monoreduced 1,3-diisopropylimidazoline-2,4,5-trithione ligands—namely, [Pt(iPr_2_timdt)_2_]—were investigated. Theoretical studies using the density functional theory (DFT) and GW-BSE approximation verified the effects of the geometry of the isopropyl moieties on the NIR absorption spectra. The calculated absorption spectra showed excellent correspondence with the experimental results. The geometry of the isopropyl groups considerably influenced the electronic structures of the metal complexes, which altered the absorption profiles of the respective geometries, as demonstrated in this research.

## 1. Introduction

The interest in metal complexes has surged in the past few decades [1,2,3] because of the potential applicability of these complexes in superconductors [4,5], photoconductors [6,7,8,9], magnetics [10,11], and linear and nonlinear optical devices [12,13,14], among other applications. As small-molecule probes for bioimaging, heavy-metal complexes offer many advantages, such as a high luminescence efficiency as well as tunable excitation and emission, owing to the modular nature of the complexes. The rich and complicated structures of the excited states of transition metals with d^6^, d^8^, and d^10^ electronic structures also give rise to excited-state phenomena such as metal–ligand charge transfer (MLCT), intraligand charge transfer (ILCT), and ligand–ligand charge transfer (LLCT). These phenomena cause the optical absorption to shift deep into the near-infrared (NIR) region [15,16], which is highly advantageous for bioimaging, as NIR radiation undergoes low scattering, has a high photon penetration rate, and does not experience interference from the autofluorescence of biological samples [13,17,18].

Complexes of bis(1,2-dithiolene) with metal ions possessing d^8^ electronic structures—for example, Ni(II), Pd(II), and Pt(II)—present peculiar properties [3,19,20,21], such as high molecular planarity and high optical absorption rates. Because of the redox non-innocence nature of the 1,2-dithiolene ligands [22,23], it is challenging to determine the charge of ligands with kernel metal atoms (M^2+^). For neutral complexes, the formal charge of the metal atom can vary between 0 and +4 when the ligand is neutral, monoanionic, or dianionic (Figure 1). These possible schemes indicate a significant degree of π–electron delocalization at the center of the molecule, constituting a metalloaromaticity relationship involving the metal atoms and ligands [24,25]. On the other hand, spectroscopic and theoretical studies of various neutral and charged species of bis(1,2-dithiolene) d^8^ metal complexes suggest that the metal atom is better described as M(II), and any redox activity occurs only in the ligand [20,22]. Hence, with their non-innocent nature, neutral complexes can be described as diamagnetic singlet species formed from two monoanionic ligands coupled in an antiferromagnetic fashion, e.g., [M(II)(L*^−^)_2_] [20]. Consequently, open-shell (and spin-non-restricted) calculations are required to reliably evaluate the ground state of these complexes [26,27]. Ultraviolet (UV)–Vis–NIR spectroscopy analyses have demonstrated that these neutral complexes show peculiar optical properties, with an intense absorption peak in the region beyond 800 nm [21]. This absorption band is directly derived from the π–π* one-electron excitation [19] between the energy states of the highest occupied molecular orbital (HOMO) and lowest unoccupied molecular orbital (LUMO). To shift the absorption peaks deeper into the NIR region, one typical strategy involves modifying the functional groups directly adjacent to the ligands to alter their radical strength [28].

The [M(R_2_timdt)_2_] neutral complex (R_2_timdt− = monoreduced 1,3-disubstituted imidazoline-2,4,5-trithione; M = Ni, Pd, Pt), as a species derived from 1,2-thiolene ligands, has been investigated over the past few decades. Neutral [M(R_2_timdt)_2_] complexes show a strikingly intense absorption at ~1000 nm (where the molar extinction coefficient is as high as 120,000 M^−1^ cm^−1^ in toluene) [21]. Since the initial studies, research on Pt and isopropyl as a metal kernel and the ligand-substituted functional group, respectively, has been scarce.

Early research focused extensively on Ni atoms, whereas the isopropyl group has rarely been studied due to its bulky nature, which makes the process of crystallization difficult [30]. However, for bioimaging, the bulkiness of isopropyl can minimize the effect of the solvent; furthermore, Pt atoms are inert in biological environments.

This study presents a theoretical approach for predicting the UV–Vis–NIR absorption spectra of [Pt(iPr_2_timdt)_2_], intending to evaluate the contribution of the geometry of the isopropyl functional group to the low-energy excitation response.

## 2. Materials and Methods

### 2.1. Methodology Section

Bigoli et al. [30] and Mogesa et al. [20] showed that as its main structure, [M(iPr_2_timdt)_2_] adopts a planar conformation with the dithiolene ligands chelating the metal atom in a square planar geometry, while the functional groups bonded to the N atoms on the ligand remain perpendicular to the body of the complex. For the evaluated complexes, the isopropyl group had a V shape; hence, the symmetry plane was between the two methyl groups. Therefore, the lowest potential of [Pt(iPr_2_timdt)_2_] was reached when the symmetry plane of the isopropyl group aligned with the planar plane of the molecule. Based on these observations, the dithiolene ligands have two possible relative configurations: *cis* and *trans*. The *cis* configuration has two distinct forms: one moves inward, while the other moves outward relative to the metal kernel. The *trans* configuration, owing to its nature, creates a chiral structure of [Pt(iPr_2_timdt)_2_] from a longitudinal flip symmetry.

Hence, six possible structures of the complex molecule are plausible, which could be split into three groups based on the combination of configurations of the groups, namely, Cx, Tx, and Mx (x = 1, 2), consisting only of *trans*, only of cis, and of a mix of both configurations, respectively. The replicate structures obtained using a Newman-like projection, along with molecular representations of [Pt(iPr_2_timdt)_2_], are illustrated in Figure 1.

To investigate the absorption spectra of the various [Pt(iPr_2_timdt)_2_] configurations, all calculations were initiated using the density functional theory (DFT) [31] framework in the VASP package [32,33]. To overcome the limitations of pure DFT calculations at the long-wavelength limit, a correction using the many-body perturbation theory of the Hedin equation was added, generally known as the GW approximation [34]. To obtain the optical absorption spectra of the molecule, the dielectric function of the system was calculated, and the theoretical extinction coefficient was extracted from the response function with the excitonic effects using the Bethe–Salpeter equation (BSE) [35,36]. The equation proved to be effective in describing the response of the structure to radiative excitation [37].

The molecule was isolated in a 20 × 20 × 20 Å cubic cell to ensure that the cell had sufficient vacuum distance to prevent overlap of the wavefunctions. All calculations were performed with the local density approximation (LDA) functional at the gamma point, using the Gaussian smearing method and a small sigma value of 0.05. Despite having drawbacks such as strong binding and overestimating the bandgap, the LDA functional provides a relatively good description of the general structure of the molecules, and is a good starting point for GW calculations. The kinetic energy cutoff was chosen to be 450 eV, which yielded a good convergence rate of under 10^−4^ eV/Å. The potential of the molecules was constructed from a set of plane-wave pseudopotentials using the projector augmented wave (PAW) method [38,39]. The structure optimizations converged when all of the calculated forces were under the threshold of 0.02 eV/Å, and the algorithm suggested no further ionic optimization steps. Based on a prior investigation, bis(1,2-dithiolene) complexes of [M(R_2_timdt)_2_] were found to generally display a diradical character, which requires an open-shell singlet model to fully describe the characteristics of the complex electronic structure in the excited states [29]. The spin-non-polarized configuration only provides a closed-shell singlet model of the molecular excitation. Therefore, the wavefunctions of the molecules obtained from geometric optimization were refined with a spin-polarized configuration for later calculations.

For the GW calculation, only the first correction with no self-consistent scheme for the independent particle Green function “G” and Coulomb-screened potential “W” (G_0_W_0_ method) was performed, with an energy cutoff of the dielectric function at 50 eV. Eighty points were considered to effectively sample the imaginary frequency from the response function. In the BSE scheme, a matrix consisting of 4 occupied bands and 13 unoccupied bands was used to describe the influence of electron–hole relations on the optical absorption spectra. A model beyond the Tamm–Dancoff approximation (TDA) was adopted [40], accounting for coupling between the positive and negative frequencies in the response function to achieve a better theoretical estimation of the frequency-dependent dielectric function. Finally, the absorption coefficients were calculated from the dielectric function with Maxwell’s identity [41] using Equation (1):(1)$$\varepsilon =\varepsilon^{\prime }+i\varepsilon^{\prime \prime }={\left(n+ik\right)}^{2}$$
where *ε* is the complex expression of the dielectric function (relative permittivity), while *n* and *k* are the refractive index and absorption coefficient, respectively.

### 2.2. Experimental Section

All solvents and reagents were of the Aldrich quality, and were used as obtained. Distilled water was used for all experiments. All reactions were monitored by performing thin-layer chromatography. Merck silica gel sheets (silica gel 60 F254, Rahway, NJ, USA) were used for the diagnostic TLC. Nuclear magnetic resonance spectra (^1^H-NMR) were collected using a Bruker AVANCE II 500 MHz (Billerica, MA, USA). Liquid chromatography/mass spectrometry analyses were performed on a Waters ACQUITY H-Class Liquid Chromatograph/SQD2 Mass Spectrometer (Milford, MA, USA). Element analyses were collected on a Thermo Scientific FLASH EA-2000 Organic Elemental Analyzer (Waltham, MA, USA). UV–Vis–NIR spectra were recorded using a 1 cm path length cell with a SHIMADZU UV-2600 UV–Vis spectrophotometer (Kyoto, Japan) at room temperature.

#### Synthesis of [Pt(iPr_2_timdt)_2_]

1,3-Diisopropylimidazolidine-2-thione-4,5-dione was synthesized via a method previously reported in [42,43]. A 25 mL mixture of 1,3-diisopropylthiourea (10 g, 62 mmol) and 2 M oxalyl chloride (in dichloromethane) in toluene (40 mL) was refluxed at 130 °C for 8 h (TLC Hex:EA = 3:1). The mixture was concentrated with a rotary evaporator under reduced pressure and recrystallized from methanol (yellow solid, 7.5 g (35 mmol), Yield: 56.5%)

[Pt(iPr_2_timdt)_2_]-Green was synthesized according to the procedure previously described in [44,45]. A suspension of 1,3-diisopropylimidazolidine-2-thione-4,5-dione (0.5 g, 2.3 mmol), Lawesson’s reagent (0.94 g, 2.3 mmol), platinum(II) chloride (0.31 g, 1.16 mmol), and toluene (20 mL) was refluxed (130 °C) for 3 h. The mixture was concentrated with a rotary evaporator under reduced pressure to dryness, and then recrystallized from dichloromethane/ethanol. It was dried under vacuum to yield a dark-green solid (0.75 g (1.1 mmol), 95.6% yield). Anal. Calcd for C_18_H_28_N_4_PtS_6_: C, 31.43; H, 4.10; N, 8.14; S, 27.96 Found: C, 33.20; H, 4.10; N, 6.95; S, 26.46, LC-MS(m/e): 687.9, H1 NMR (δ ppm: CDCl_3_) 5.584 (4H, m), 1.717 (24H, s).

[Pt(iPr_2_timdt)_2_]-Blue; 1,3-diisopropylimidazolidine-2-thione-4,5-dione (0.5 g, 2.3 mmol) was dissolved at 80 °C in toluene (20 mL). Then, Lawesson’s reagent (0.94 g, 2.3 mmol) was added to the toluene mixture at 80 °C. The reaction mixture was then refluxed for 5 min after an equivalent amount of platinum(II) chloride (0.31 g, 1.16 mmol) was added and refluxed for 3 h. The resulting mixture was cooled by standing in a refrigerator (4 °C) for 4 h. A bluish precipitate was filtered and washed with cold toluene and petroleum ether (20 mL). It was then dried under vacuum to yield a dark-blue crystalline solid (0.42 g (0.61 mmol), 53 % yield). Anal. Calcd for C_18_H_28_N_4_PtS_6_: C, 31.43; H, 4.10; N, 8.14; S, 27.96 Found: C, 30.19; H, 3.93; N, 7.75; S, 26.87, LC-MS(m/e): 687.2, H1 NMR (δ ppm: CDCl_3_) 5.584 (4H, m), 1.717 (24H, s).

## 3. Results and Discussion

### 3.1. Equilibrated Structures

Owing to the crystallization procedure, the configuration of isopropyl groups in the nickel metal complex of this molecular species was proposed based on the single-crystal X-ray structure diffractometry (XRD) data to have a *cis* form on the ligand, staying perpendicular to the molecular geometric plane, and always facing outward [20,30]. This configuration is aligned with the C1 structure that we introduce in the methodology section. For the calculation of equilibrated structures, it reached convergence for the structure with minimal distortion in the planarity of the molecules, where the appended isopropyl functional groups mirrored the planar form of the complexes, as shown in Figure 2.

Due to the large van der Waals radii of the isopropyl functional groups and the S atoms, notable changes in bonding and angles produced by the steric interaction were expected, and such changes are presented in Table 1. The illustration on the notion of those bonds and angles are presented in Figure 3. For the bonds and angles relating to the isopropyl functional groups, the average values of the C–N bond and θ_3_ angle show a relatively large change from each structure—as large as 1.464–1.470 Å and 123.2–127.2° in Cx structures, respectively. There are also noticeable changes in the values involved in the interaction between Pt atoms and ligands, which are C–S and Pt–S bonds, and θ_1_ and θ_2_ angles, and the largest variations are observed in Cx structures. This behavior shows a great influence of the isopropyl group’s isomerism on the geometry of the metal complexes. On the other hand, those values of bonds and angles mentioned earlier in Tx structures have the median values among two forms of Cx and Mx geometries (1.702 Å, 2.269 Å, 1.466 Å, 88.0°, 100.9°, 125.1° for C–S, Pt–S, and C–N bonds and θ_1_, θ_2_, and θ_3_ angles, respectively), showing that the *trans* isomer mediates the steric interaction between isopropyl functional groups and S atoms. Hence, we expect **T1** and **T2** to be thermochemically favorable, particularly in the solvated model. To differentiate the effects of geometric isomerism on the metal complex—especially between **T1** and **T2** structures—the deviation from average in the θ_1_ angle is the only noticeable value (0.2°), which indicates an asymmetric widest dimension of **T2** compared to the **T1** structure. The molecular structures of **T1** and **T2** are shown in Figure 2.

### 3.2. Optical Properties

Figure 4 shows the optical absorption spectra of the multiple configurations of [Pt(iPr_2_timdt)_2_], obtained from theoretical calculations using the LDA/GW-BSE method. The spectra of the **T1** and **T2** structures each showed a strong absorption peak at 1017 nm, which differed slightly in intensity. For the **T1** structure, the absorption peak was relatively less intense (0.86 × 10^5^ a.u.) than that of the **T2** configuration (1.05 × 10^5^ a.u.). As for the spectroscopic measurements, an intensive absorption peak at 1013 nm was observed for **Pt-DT-IS-S2** (1.135 × 10^5^ a.u.), and **Pt-DT-IS-S1** produced a less intensive peak (0.71 × 10^5^ a.u.) at the same wavelength. The experimental UV–Vis–NIR absorption spectra and the calculated spectra showed a relative equivalence between **Pt-DT-IS-S1** and the **T1** structure (and between **Pt-DT-IS-S2** and the **T2** structure), with the consistency represented in Figure 5.

Other configurations, such as those comprising only *cis* isomers of the ligands (**C1** and **C2** structures) or a mix of isomers of the ligands (**M1** and **M2** structures), showed a set of optical absorption peaks spread over the visible wavelength region, with no definitive trend. The **C1** structure had the most intense absorption peak at 1009.3 nm, and a less intense secondary peak at approximately 880 nm. In contrast, for **C2**, the most intense absorption peak shifted back to 504.7 nm, with a low intensity of 0.27 (10^5^ a.u.), which is much lower than those of **C1**, **T1**, and **T2** (0.83, 0.86, and 1.05, respectively). The **M1** structure had an intense absorption peak with a magnitude of 1.03 (10^5^ a.u.) at 761.2 nm. In contrast, the **M2** structure showed several recognizable peaks spanning a wide bandwidth range from 500 nm to 1200 nm. The most intense peak was centered at 794.3 nm, with an intensity of 0.44 (10^5^ a.u.), while other less intense peaks were observed around 550, 900, and 1150 nm.

### 3.3. Discussions

Platinum metal complexes of dithiolene species have been confirmed to possess an intermediate diradical character (y ≈ 0.3) [46], which significantly contributes to the charge transfer states of the molecules, leading to the appearance of intense peaks in the NIR region. This phenomenon originates from the non-innocence of the dithiolene ligands, while the d^8^ orbital of the Pt(II) atom contributes to the metalloaromaticity in the center of the complexes, forming a bridge between the radicals on the ligands. Changing the configuration of the isopropyl functional groups in the [Pt(iPr_2_timdt)_2_] geometries creates a different set of interactions between isopropyl groups and S atoms, inducing the modification of the bond lengths and bond angles of those atoms related to the central square-planar geometry. This leads to an imbalance between two ligands, which influences how the radicals on the ligand interact with one another through within-bond interaction. The shift in the nature of the diradical coupling character also changes the transition moments between the ground and excited states, or even allows previously forbidden transitions between the HOMO and LUMO states, leading to new peaks in the visible region, as shown in Figure 4. In the absorption spectra of the **T1** and **T2** structures, the *trans* configuration of the isopropyl functional groups on the ligands assists in minimizing the total interactions between the S atoms and isopropyl groups, which leads to the least internal tension in the molecule. The difference in the absorption intensity can be attributed to the spatial orientation of the *trans* configuration of the ligand, creating an expansion/contraction of the θ_1_ angle along the widest dimension of the molecule in the **T2** structure compared to the **T1** structure.

## 4. Conclusions

[Pt(iPr_2_timdt)_2_] was synthesized, and its optical properties were thoroughly investigated by spectroscopic measurements and theoretical calculations. The synthesized complexes exhibited the peculiar property of having two different stable structures with different optical absorption patterns. Calculation of the UV–Vis–NIR spectra using the LDA functional, along with correction using the GW-BSE method, verified the influence of the configuration of the isopropyl functional group on the optical absorption spectra. Considering the high diradical character of bis(1,2-dithiolene) metal complexes, these results suggest an indirect contribution of the geometries to the electronic properties of this species of molecules, which needs to be considered when investigating this species of metal complexes.

## Data Availability

The data presented in this study are available on request from the corresponding author.

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
