# Peer review of "Near-Infrared Absorption Properties of Neutral Bis(1,2-dithiolene) Platinum(II) Complexes Using Density Functional Theory"

_nanomaterials, 2022, doi:10.3390/nano12101704_

Round 1
Reviewer 1 Report
This manuscript describes absorption properties of neutral bis(1,2-dithiolene)platinum complexes based on DFT calculations. The presentaion is simple and concise except for some issues indicated below. This reviewer recommend possible publication of this manuscript after the authors reply to the following comments with necessary revision of the manuscript. (1) P. 1, L. 31, photoconductors [6,7] --> some important papers are missing, for example, JACS, 134, 18656; Adv. Mater. 24, 6153. (2) P. 1, L. 42, d6 --> d8? (3) P. 4, L. 155, (XPS) data. [16, 26] --> More detailed and more relevant XPS data on Ni-dithiolene complexes are published in Eur. J. Inorg. Chem., 2014, 4000. (4) P. 6, L. 192-P. 7, L. 218 --> Although the section is titled as "3.2 Optical properties", the actual secion include the syntheses and measurement of the spectra of the obtained compounds. At least, the details of sythetic procedure and spectroscopic measurements should be described in "2. Materials and Methods" in P. 2. In addition, the long abbreviations of the compounds such as Pt-DT-IS-S1 should be shortened, if possible. (5) P. 7, L. 237-P. 8, L. 267, "Conclusions" --> The content includes too many details and thus rather like short discussion than conclusion. Additionally, some logics are incomprehensive, for example, the relationship between the center of mass of the ligands and diradical character (L. 246-L. 250), and what "theoretical measurements (L. 260)" is meant for. As a result, main concluison might be misunderstood by some readers, and even the central point of view of this manuscript, which is presented in the title and introduction, appears to be inconsistent with this "conclusions". (6) P. 7, L. 242, d6 orbital --> d8? Anyway, the electronic configuration and the name of orbital should not be mixed, which may be misleading. (7) There are some senetences where the usage of English words is not standard. For example, P. 5, L. 171, "In another hand, ....".
Author Response
We thank reviewer 1 for a careful review of our paper and helpful suggestions. Please see our responses to your comments underneath each point. We believe these very thorough comments and responses to the comments have made the paper stronger and more useful. We do truly appreciate the time and effort of reviewer 1 again. Please see the attachment for detail report.

Reviewer 2 Report
The materials for infrared absorption or luminescence shown in this report are needed not only in the biological field but also in energy and sensors. I think ths report is enough to accept after minor rivision.
You should check superscrips or subscript for some words .
p2L74 "120.000 M-1 cm-1", p7L222 "1.0x10-5 M"
Author Response
We thank reviewer 2 for a careful review of our paper and helpful suggestions. Please see our responses to your comments underneath each point. We believe these very thorough comments and responses to the comments have made the paper stronger and more useful. We do truly appreciate the time and effort of reviewer 2 again. For detail revision, please see the attachment.

Reviewer 3 Report
In this contribution, the authors investigate an applicability of the theoretical methods for evaluation of the effect of the geometry of the isopropyl moieties in Pt(II) bis(1,2-dithiolene) complexes on their NIR absorption spectra. For this purpose, absorption spectra of the different conformers of [Pt(iPr2timdt)2] complex has been computed and compared with experimental spectrum. It has been found that “the geometry of the isopropyl groups considerably influenced the electronic structures of the metal complexes”. The manuscript itself is well-written and organized. In my opinion, the reviewed work is high quality and of the proper impact/scope for this journal. Thus, I recommend this paper be published in this journal after minor revision to address the following concerns:
- The sentence “These phenomena cause the optical absorption to shift deep into the near-infrared (NIR) region, …” is recommended to supply by recent relevant references such as: 10.1021/acs.inorgchem.5b02933, 10.1021/acs.jpclett.0c01358, 10.1016/j.poly.2021.115484, 10.1039/D1DT04325K.
- The choice of the isopropyl group in the ligand is not so obvious. Why exactly isopropyl, and not, for example, tert-butyl?
- The procedure for synthesis of the known complexes Pt-DT-IS-S1 and Pt-DT-IS-S1 should be moved from the results and discussion part to the Experimental part.
- Please indicate the valence state of platinum in the title of the paper.
Author Response
We thank reviewer 3 for a careful review of our paper and helpful suggestions. Please see our responses to your comments underneath each point. We believe these very thorough comments and responses to the comments have made the paper stronger and more useful. We do truly appreciate the time and effort of reviewer 3 again. Please see the attachment file for detail revision.
